# Air Emissions from Natural Gas Facilities in New York State

**DOI:** 10.3390/ijerph16091591

**Published:** 2019-05-07

**Authors:** Pasquale N. Russo, David O. Carpenter

**Affiliations:** Institute for Health and the Environment, University at Albany, 5 University Place, Rensselaer, NY 12144, USA; pnrusso@albany.edu

**Keywords:** Compressor stations, natural gas emissions, fracking, pipelines, cancer

## Abstract

While New York has banned fracking, new and expanded natural gas pipelines are being constructed across the state. Our previous studies have reported that compressor stations are a major source of air pollution at fracking sites. We have used two federal datasets, the U.S. Environmental Protection Agency’s (EPA) National Emissions Inventory and Greenhouse Gas Inventory, to determine what is known concerning emissions from the compressor stations along natural gas pipelines in the state. From a total of 74 compressor stations only 18 report to EPA on emissions. In the seven year period between 2008 and 2014 they released a total of 36.99 million pounds of air pollutants, not including CO_2_ and methane. This included emissions of 39 chemicals known to be human carcinogens. There was in addition 6.1 billion pounds of greenhouse gases release from ten stations in a single year. These data clearly underestimate the total releases from the state’s natural gas transportation and distribution system. However, they demonstrate significant releases of air pollutants, some of which are known to cause human disease. In addition, they release large amounts of greenhouse gases that contribute to climate change.

## 1. Introduction

The World Health Organization recently stated that there are 4.2 million deaths every year as a result of exposure to ambient outdoor air pollution [1]. While there are natural sources of air pollution, most comes from combustion of fossil fuels, such as wood, coal, animal dung, oil, gasoline, and natural gas.

In addition to the dangers to human health posed by air pollution, combustion of fossil fuels generates greenhouse gases, especially carbon dioxide [2]. Carbon dioxide levels are higher at present than they have been since human life appeared and most is due to human activity. Methane, the major component of natural gas, is also a potent greenhouse gas, and most comes from fossil fuel exploitation and agriculture. It is actually more potent as a greenhouse gas than carbon dioxide, although its residence time in the atmosphere is shorter [3]. 

With the development of hydrofracturing (“fracking”) there has been a sudden abundance of natural gas. Natural gas is a cleaner fuel than coal when burned, but still a fossil fuel that, when burned, generates a variety of air pollutants, including particulates, volatile organic compounds and greenhouse gases. 

In 2014 Macey et al. [4] published a study of emissions from fracking wells and associated compressor stations in five states, and showed that there were releases of air pollutants that exceeded federal standards, sometimes by as much as 10,000-fold. The major exceedances were for formaldehyde, hydrogen sulfide, benzene, and other volatiles. On the basis of this and other studies New York State banned fracking. There is convincing evidence for elevations in a number of diseases in areas where fracking is ongoing, including elevations in cancer [5], birth defects [6], preterm birth [7], upper respiratory symptoms [8], exacerbations of asthma [9], and increased headaches and fatigue [10].

Any relief felt by New Yorkers because of the ban on fracking was, however, not totally justified. In nearby Pennsylvania fracking was occurring at a frenetic pace. In order to access markets for Pennsylvania natural gas there was a sudden construction of new pipelines and expansion of existing pipelines across New York to bring Pennsylvania natural gas to New England and ports for overseas shipment. 

The components at fracking sites that were most associated with exceedances in air pollution standards in the Macey et al. study [4] were, more often than not, the compressor stations. Because compressor stations are critical to the operation of a natural gas pipeline, in order to push the gas through, we began a study of emissions from the compressor stations along pipelines in New York. 

## 2. Materials and Methods

Data was obtained from three federal datasets, the USEPA (The United States Environmental Protection Agency) National Emissions Inventory (NEI), the USEPA Greenhouse Gas Inventory (GHGI) and the Homeland Security facilities list. In addition, we reviewed the New York Department of Environmental Conservation’s (DEC) list of facilities, which were granted Title V permits and those granted Air State Facility permits. 

The NEI reports air emissions in four categories (point, non-point, road, and non-road). Emissions from compressor stations are considered to be point sources. The NEI reports releases in pounds and reports are released only every third year. In this publication we use information released in 2008, 2011, and 2014. Only those point sources that are designed as a Title V facility are included in the NEI for New York. A Title V permit is required by the DEC for any point source facility deemed to be “major” under department regulations. Those facilities holding Title V permits are required to have pollution control measures in place and to report amounts of emissions to the DEC on a regular basis. 

There are a total of 316 chemicals reported in the NEI, but the NEI does not include CO_2_ or methane for point sources such as compressor stations. Therefore, information on these gases is available only from the GHGI. Some volatile compounds are specifically listed in the NEI, such as benzene, hexane, toluene, xylenes, and there is a separate listing for VOCs. Particulates are listed in two categories: PM_10_ and PM_2.5_ (PM: Particulate Matter). 

The essential components of New York’s natural gas transportation and distribution system includes: 49,126 miles of “distribution” pipelines, 4562 miles of “transmission pipelines”, 74 natural gas compressor stations, 26 underground storage facilities with 246 billion cubic feet (bcf) of working gas capacity (2.7% of the U.S. total), 3 peakshaving liquid natural gas (LNG) facilities, and 79 natural gas powered electric utilities (Standard Industrial Classification code 4911).

Table 1 shows the components of the natural gas distribution system in New York that we have been able to document. Upon examination of the five federal and state datasets, we found 82 compressor stations in New York. Of these, 21 hold a Title V permit, and 30 hold an Air State Facility permit. There are an additional 31 compressor stations for which the permit type is unknown. In addition, there are 27 facilities that are a part of natural gas distribution system. We were able to obtain emissions data only from 18 of the 21 compressor stations that hold Title V permits and only ten of these compression station that report results to the GHGI. As shown in Table 1, there are also other components along existing pipelines for which no information on releases is available. 

Because well-flow and reservoir pressure decrease over time and distance, natural gas compressor stations are required to maintain or increase gas flow from natural gas fields into the pipeline system. These are generally made of two basic types: In small and medium-size fields reciprocating compressors are used. In large fields centrifugal compressors are deployed.

The GHGI provides information on six chemicals that are classified as point source of greenhouse gases. This include carbon dioxide, methane, nitrous oxide, perfluorinated compounds, hydrofluorocarbons, and sulfur hexafluoride. Only the first three are of concern in relation to compressor stations. Because nitrous oxide is also reported in the NEI we have used only the carbon dioxide and methane information from the GHGI.

## 3. Results

Figure 1 shows the locations of the natural gas pipelines and 63 of the 74 compressor stations in New York State (NYS) for which we have geospatial coordinates. 

Table 2 lists the total pounds of chemicals released from the 18 compressor stations that report emissions to the NEI for each of three reporting years. By far, the largest releases are for nitrogen oxides and carbon monoxide, followed by volatile organic compounds, formaldehyde and PM_10_. The NEI listing for volatile organic compounds includes some individual compounds that are also listed separately. The table shows both total volatiles and the specific amounts of three individual compounds that are classified as known human carcinogens (benzene, formaldehyde, 1,3-butadiene). The seven year total (2008–2014) was calculated by determining the average chemical release for three years and multiplying that sum by seven.

Table 3 shows the NEI releases from individual compressor stations that hold a Title V permit, listed by town. The magnitude of releases varies considerably by station and in some cases also by year. Some of the variations may be explained by changes in equipment deployed at the stations.

Table 4 shows emissions of CO_2_ and methane from ten compressor stations reported in the GHGI in 2014. There were no data for eight of the 18 compressor stations holding Title V permits. The total release of greenhouse gases for this single year totaled more than six billion pounds for these ten compressor stations alone.

Table 5 list those chemicals released from compressor stations in New York that are known to be human carcinogens and lists the authorities or agencies that have made that classification.

For each chemical the IARC monograph includes a list of human cancers that are known to be caused by that chemical, as well as cancers for which there is a “positive association”. For the known carcinogens released by New York compressor stations, IARC concludes: Arsenic is known to cause human lung, urinary bladder, and skin cancer, and there is positive indications for kidney, liver, and prostate cancer. Benzene is known to cause acute myeloid leukemia, and there are positive associations for non-Hodgkin lymphoma, chronic lymphoid leukemia, multiple myeloma, chronic myeloid, leukemia, acute myeloid leukemia in children and lung cancer. Beryllium is a known cause of lung cancer. Cadmium is a known lung carcinogen, and there are positive associations for kidney and prostate cancers. Formaldehyde is a known cause of human nasopharynx cancer. Nickel is known to cause nasal cavity and paranasal sinus cancers. Liver and bile duct cancers are known to be caused by exposed to vinyl chloride.

## 4. Discussion

These results show that there are significant emissions of criteria air pollutants (CAPs), hazardous air pollutants (HAPs) and greenhouse gases from compressor stations in New York, based on information contained in two federal datasets. Clearly the results reported are significant under-estimations of total releases from compressor stations in New York. There is, for example, no information on air pollutants for the 52 non-Title V compressor stations in the NEI or the compressor stations at the state’s 23 natural gas storage sites. The EPA’s GHGI only contains entries for 15 of the state’s 72 compressor stations. Nevertheless, the available results show significant emissions, especially of carbon dioxide, methane, nitrogen oxides, carbon monoxide, formaldehyde, and particulates.

The combination of air pollution and climate change pose serious threats not only to human health but also to life as we know it. Our global energy sources have been primarily combustion of fossil fuels. Clearly coal combustion is among the worst in terms of release of both carbon dioxide and particulates. With new technologies that allow cost-effective extraction on natural gas it has been possible to phase out many of the older coal-fired power plants, as well as to replace coal and oil with natural gas for other domestic uses. However natural gas is still not a “clean” fuel. It still produces greenhouse gases, particulates, and volatile organic compounds when combusted.

However, it is not only the combustion of natural gas that poses threats to the environment. There is a significant leakage of methane from the natural gas systems [11,12]. Alvarez et al. [13] used facility measurements and aircraft observations to conclude that about 1.2% of methane escapes during extraction and use, while the study of Karion et al. [11] found that between 6.2 and 11.7% of the extracted natural gas in one month for one production field in Utah escaped into the air. Miller et al. [14] reported that current EPA inventories of methane emission underestimate the actual amount by about 1.5-fold, and that in at least some parts of the US, fossil fuel extraction and refining contribute about 45% of total methane releases. Howarth et al. [15] postulated that methane has a greater impact on greenhouse gas production than coal, all because of the escape of methane during extraction and distribution. This is a serious concern quite independent of the greenhouse gases produced when methane is burned. On a global basis the amount of methane released has been increasing in recent years for reasons that are uncertain [16], but it is certain that the increasing development of fracking and pipeline transport of natural gas is one component of this increase.

The health consequences of natural gas extraction and transport are of concern. Air pollution from whatever source is documented to be a risk factor for many different diseases. IARC has rated air pollution as a Group 1, known human carcinogen [17]. Even low levels of air pollution increase the risk of hospitalization for cardiovascular [18], respiratory disease [19,20], and death [21]. Recent studies document the associations between levels of air pollution and a wide variety of other health effects, including stroke [22], hypertension [23], preterm birth [24], and mild cognitive impairment in older adults [25].

While air pollution from compressor stations is only one, and certainly not the major, source of air pollution, most of these known effects of air pollution have been documented to be elevated in individuals living near fracking sites, as referenced in the Introduction. Of particular concern is cancer. McKenzie et al. [5] reported an odds ratio of 4.3 (95% CI, 1.1–16) for development of acute lymphocytic leukemia in children ages 5-24 who lived near sites of oil and gas development. Finkel [26] found that there were statistically significant elevations in bladder and thyroid cancer in Pennsylvania counties with fracking, as compared to those without fracking. The results in Table 5 show that a number of the components monitored from compressor stations in NYS are known to be human carcinogens. Of particular concern with regard to leukemia is the release of benzene, which has long been known to increase the risk of leukemia, even at low levels of exposure [27]. We have previously reported that hospitalization for leukemia was significantly elevated among individuals who lived in a zip code with a hazardous waste site containing benzene [28]. And benzene is not the only carcinogen being released from compressor stations (Table 5). These and other studies show that even low levels of exposure over periods of time can increase the risk of cancer and other diseases.

## 5. Conclusions

In spite of the positive features of natural gas over other forms of fossil fuels, extraction, distribution, and combustion of natural gas generates air pollution and greenhouse gases. Our results add to the evidence for urgency for the replacement of fossil fuels with renewable sources of energy in order to both protect human health and reduce the immediate and long-term threats arising from climate change.

## Figures and Tables

**Figure 1 ijerph-16-01591-f001:**
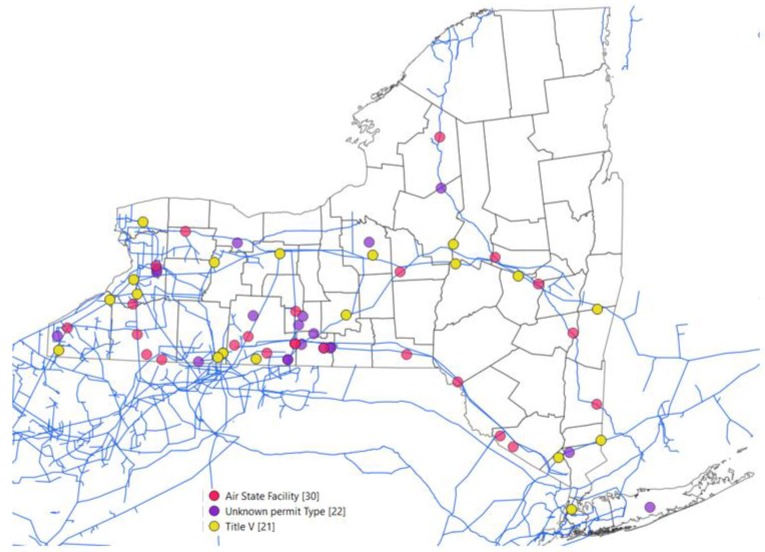
Natural gas pipelines across New York State (blue lines) and associated compression stations. Compressor stations are indicated by colored circles, indicating different types of permit. The black lines represent county and state boundaries.

**Table 1 ijerph-16-01591-t001:** New York State (NYS) compressor stations and related facilities.

Compressor Stations by Permit Type	74
Title V	21
Air State Permit	30
No permit information	22
Mapped in Figure 1	63
Other Facilities	
Underground storage facilities	26

**Table 2 ijerph-16-01591-t002:** The United States Environmental Protection Agency (USEPA) National Emissions Inventory (NEI) emissions (pounds) by chemical: 2008–2014.

Rank	Chemical	NEI-2008	NEI 2011	NEI 2014	2008–2014 Estimate
1	Nitrogen oxides	2,269,341	2,991,946	2,487,284	18,079,997
2	Carbon monoxide	1,415,996	2,029,497	1,850,403	12,357,089
3	Volatile organic compounds	374,277	830,863	902,548	4,917,940
	Benzene	110,334	229,882	220,928	1,309,335
	Formaldehyde	2029	3875	3199	21,240
	1,3-Butadiene	273	999	751	4719
	Other VOCs	261,642	596,107	677,671	3,582,646
4	PM 10 Primary (Filt + Cond)	107,946	241,483	189,665	1,257,888
5	Sulfur dioxide	7587	13,894	58,287	186,124
6	Ammonia	262	209	174	1505
7	Nickel	169	21	107	692
8	Manganese	104	0	47	350
9	Mercury	17	6	6	70
10	Chromium III	16	0	7	56
11	Phenanthrene	4	14	2	48
12	PAH, total	0	0	15	35
13	Cadmium	9	0	4	30
14	Fluorene	2	8	1	28
15	Benz[a]anthracene	4	2	2	19
16	Fluoranthene	2	2	1	11
17	Anthracene	0	4	0	10
	Other chemicals (19)	4	4	1	21
	Total	4,175,740	6,107,954	5,488,555	36,801,914

**Table 3 ijerph-16-01591-t003:** USEPA NEI emissions (pounds) by town: 2008–2014.

Town	NEI-2008	NEI 2011	NEI 2014	2008–2014 Estimate
Andover	110,115	194,987	241,599	1,275,636
Carlisle	692,603	476,313	490,580	3,872,157
Clifton Spring	288,483	434,874	167,787	2,079,336
Clymer	39,824	350,616	51,161	1,030,402
Concord	331,832	934	317,222	1,516,638
Eden 1	458,153	1,062,255	502,879	4,721,002
Eden 2	78,928	2605	115,654	460,102
Forestville	94,041	72,528	0	582,989
Ithaca	127,634	81,663	119,933	768,204
LaFayette	265,611	485,718	388,652	2,659,956
New Hartford	45,120	53,281	9667	252,161
Riders Mills	282,478	243,945	447,805	2,273,200
Southeast	156,151	220,860	273,543	1,517,959
Stony Point	234,506	236,490	310,657	1,823,858
Willing	108,133	201,357	245,720	1,295,491
Winfield	707,609	1,782,565	1,773,419	9,948,386
Woodhull	102,213	178,035	22,769	707,040
York	52,304	28,925	9,510	211,725
Total	4,175,740	6,107,954	5,488,555	36,996,244

**Table 4 ijerph-16-01591-t004:** USEPA greenhouse gas emissions (Pounds): 2014.

Town	CO_2_ non-Biological	CH_4_	Total
Andover	10,527,722	2,725,796	13,253,518
Carlisle	85,376,555	176,468	85,553,023
Clymer	9,331,054	62,399	9,393,453
Concord	40,485,642	1,789,128	42,274,770
Eden 1	32,559,592	287,408	32,847,000
Eden 2	100,509,508	127,710	100,637,218
LaFayette	79,500,361	233,158	79,733,519
Southeast	241,438,739	243,453	241,682,192
Stony Point	165,034,546	223,690	165,258,236
Winfield	100,681,909	261,740	100,943,649
Total	6,122,265,234	6,217,993	6,128,483,227

**Table 5 ijerph-16-01591-t005:** Known human carcinogens released by NYS Title-V Compressor Stations.

Chemical	Authority	Stations Reporting
IARC	CA P65	USEPA	2008	2011	2014
Acetaldehyde	2B	Known	B2	13	14	12
Arsenic	1	Known	A	5	4	2
Benz[a]anthracene	2B	Known	B2	9	7	2
Benzene	1	Known	Known/Likely	16	15	14
Benzo[a]pyrene	1	Known	B2	7	5	1
Benzo[b]fluoranthene	2B	Known	B2	9	8	3
Benzo[k]fluoranthene	2B	Known	B2	5	0	0
Beryllium	1	Known	Known/Likely	4	5	2
1,3-Butadiene	1	Known	Known	13	12	12
Cadmium	1	Known	B1	9	5	4
Carbon tetrachloride	2B	Known	Likely	6	4	4
Chloroform	2B	Known	Likely	6	4	4
Chrysene	2B	Known	B2	9	8	3
Cobalt	2B	Known	NR	5	4	2
Dibenzo[a,h]anthracene	2A	Known	NR	4	0	0
1,3-Dichloropropene	2B	Known	B	8	0	3
7,12-Dimethylbenz[a]anthracene	NR	Known	NR	0	3	2
Ethyl benzene	2B	Known	D	13	13	13
Ethyl chloride	3	Known	NR	4	4	4
Ethylene dibromide	2A	Known	Likely	6	4	5
Ethylene dichloride	2B	Known	B2	6	3	3
Ethylidene dichloride	NR	Known	C	6	3	3
Formaldehyde	1	Known	B1	18	17	16
Indeno[1,2,3-c,d]pyrene	2B	Known	B2	1	4	1
Lead	2B	Known	B2	16	8	10
3-Methylcholanthrene	NR	Known	NR	0	3	0
Methylene chloride	2A	Known	Likely	11	6	8
Naphthalene	2B	Known	C	15	15	12
Nickel	1	Known	A	11	6	5
PM_2.5_ Filterable	1	MC	NR	18	18	15
PM_2.5_ Primary (Filt + Cond)	1	MC	NR	18	18	15
PM Condensable	1	No Record	NR	18	18	15
Propylene dichloride	1	Known	NR	6	3	3
Propylene oxide	2B	Known	B2	8	5	4
Styrene	2B	Known	NR	6	4	4
1,1,2,2-Tetrachloroethane	2B	Known	Likely	10	7	6
Tetrachloroethylene	2A	Known	Likely	4	4	4
1,1,2-Trichloroethane	3	Known	C	6	4	4
Vinyl chloride	1	Known	A	6	4	5

IARC, International Agency for Research on Cancer of the World Health Organization. CA-P65, State of California Proposition 65. USEPA, Chemicals Evaluated for Carcinogenic Potential, Annual Cancer Report, 2018.IARC:1-Known, 2A-Probable, 2B-Possible, 3-Unclassifiable (evidence of carcinogenicity is inadequate in humans and inadequate or limited in experimental animals), NR-No RecordsCA P65:MC-Member candidateUSEPA:Group A: “Human Carcinogen” - There is enough evidence to conclude that it can cause cancer in humans. Group B1: “Probable Human Carcinogen” - There is limited evidence that it can cause cancer in humans, but at present it is not conclusive. Group B2: “Probable Human Carcinogen” - There is inadequate evidence that it can cause cancer in humans but at present it is far from conclusive. Group C: “Possible Human Carcinogen” - There is limited evidence that it can cause cancer in animals in the absence of human data, but at present it is not conclusive. Group D: “Not Classifiable as to Human Carcinogenicity” - There is no evidence at present that it causes cancer in humans. Group E: “Evidence of Non-Carcinogenicity for Humans” - There is strong evidence that it does not cause cancer in humans.

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
