# Peer review of "Air Emissions from Natural Gas Facilities in New York State"

_ijerph, 2019, doi:10.3390/ijerph16091591_

Round 1

Reviewer 1 Report

Merely reporting emissions estimates with no context contributes nothing to the scientific  literature.  At a minimum, the emissions estimates should be compared to other sources of pollutants to give the reader context of whether or not these are significant sources of emissions.  Saying they are significant does not necessarily mean they are.  

Further, emission rates cannot be directly translated to health risk without conversion to atmospheric concentrations.  Again, at a minimum, the emission rates should be modeled using an air dispersion model to estimate ground level concentrations of contaminants.  Only then can the emission rates be put into a health context.  EPA makes air dispersion models available for free: https://www.epa.gov/scram/air-quality-dispersion-modeling-preferred-and-recommended-models

Upwind and downwind air sampling for the chemicals of interest would provide even more useful information.  I realize air sampling is not cheap, but modeling costs nothing but time.

Author Response

While we totally agree that reporting emissions without direct measurement of air concentrations has limitations, we do not agree that it is without value.  Obviously is our previous study (Macey et al.) we had measurements of air levels of contaminants, but not only is such monitoring expensive, it could never be done systematically at all of the compressor stations in NYS.  There is value in the reporting of these emissions data because, while they are all from federal data-sets, the information is hidden in a way such that it is not easily available to anyone, including both policy makers and the public.  Certainly it is only a part of the big picture, but we believe it to be a valuable contribuion.  

We also agree that it would be ideal to do an air dispersion model.  But again this is not realistic because it would have to be done for each site.  There is only so much that can be included in a single study.  We expect to follow up with more detailed analysis in future manuscripts.

Reviewer 2 Report

Compressor stations are known to use motors powered by both natural gas and diesel.  It would be helpful to know which are which and see the analysis broken out by compressor engine fule since this could constitute much of the source of the contaminants observed.  If that information is available it would add greatly to the paper

Author Response

These are valuable comments and we have tried to address each of them.  We have removed some of the rather inappropriate language that gave the op-ed tone.  We have expanded the discussion of what compressor stations do and increased the discussion of the whole pipeline network. 

We have tried to tighten the discussion of health effects of air pollution, but it is difficult to reference that information without massively increasing the size of the manuscript.  The reviewer is certainly correct that the discussion on this is very general, and even with the revisions it is still general. 

We also agree that the information on compressor station emissions should be put into perspective with other emissions.  This is in fact a study we have ongoing and hope to submit for publication soon.  However that analysis is too large to be incorporated in this manuscript.  The data is contained within the three federal data sets we have used here, but it is massive.

It is a good question as to what NYS should do with this information.  While the data we have presented all came from federal data sets, it is hidden there is a way that is not readily available, and  even those who collect that data make little effort to summarize it and use it.  So we believe that the first step is to make the data readily available to both policy makers and the public.

Reviewer 3 Report

Summary

The authors characterize 82 compressor stations in NY state that operated between 2008-2014 in terms of location and air emissions. They discuss potential health consequences of this natural gas infrastructure related to air pollution and climate change. The manuscript is informative, but I have a few suggestions and question below.

Major

1.     The manuscript takes an informal or op-ed tone at points. There is no reason to state multiple times in the discussion that “clearly” something is true. Another example is the introduction where the authors write: “As a result of this ban many New Yorkers breathed a sigh of relief.” Please just state the facts objectively and let readers draw their conclusions.

2.     As the manuscript focuses on compressor stations, a bit of background on these stations should be added. What do compressor stations do? Why does NY have so many? Are they infrastructure for conventional natural gas extraction? Are they new or have they been in place for some time? Who operates them? 

3.     Beginning line 150: the health consequences paragraph is extremely broad. The authors state that air pollution is bad for health. While correct, a more specific discussion of particular air pollutants of concern related to compressor stations and fracking would be more useful.

4.     Adding some context to the amount of emissions would be helpful. I went on the USEPA website and looked at New York state emission trends by pollutant. You could state the percentage of emissions from compressor stations among point sources of pollution or from overall state emissions. 

5.     Table 5: any information on the volume of these released? Were these pollutants measured at all compressors? As it stands, this table makes it seem like each of these pollutants were found at every compressor station. Allowing the reader to understand something about frequency and “dose” would be helpful.

6.     The conclusion is extremely general. While I agree with what the authors have written, for this manuscript it would be more useful to focus on compressor stations. What should NY state do about them? Are there alternatives? Perhaps again quantify here the concerns related to compressors and the proportion of emissions they represent of point sources or natural gas infrastructure as a whole. In 2014, it looks like the compressor stations (per data in Table 2) represented 0.2% of statewide stationary source overall VOC emissions, but 7.2% of benzene emissions and 1.2% of nitrogen oxide emissions.

Minor

1.     The EPA reports that the U.S. emitted 15.1 trillion pounds of CO2 equivalent greenhouse gas emissions in 2014. Assuming that between 2008-2014 emissions were constant, the authors estimate that just 10 compressor stations in NY state accounted for (6.1/14/1000/15.1*100) 0.003% of nationwide emissions in 2014. That’s impressive. If all 82 had a similar level of emission this would be 0.02% of nationwide emissions.

2.     Small typo in lines 20-21 of abstract: “In addition, they release a large amount of...” 

3.     Line 30-31: this sentence could use a citation, perhaps the IPCC report 

4.     Lines 35-36 could benefit from a citation, perhaps from the US EIA.

5.     Line 36: clarify what you mean by: “Natural gas is a cleaner fuel than coal.” When it burns? The whole process? Do you need to make a comparison to coal to make your point? This may open up a whole debate about coal vs. natural gas that you do not want to derail the main message of the manuscript.

6.     What does this mean? “When making that announcement the Commission of Health 42 waived our publication.” Also, please cite your study.

7.     I would not characterize the health literature as “elevations” as many of these studies did not have whole population data. I would rather state that these studies have found associations between residential proximity to unconventional natural gas development and adverse health outcomes. 

8.     Need to define USEPA at first use. 

9.     Lines 80-81: check sentence for clarity/typo. It’s difficult to follow.

10.  Please add a scale bar to Figure 1 for those less familiar with NY state.

11.  Line 92, there is an extra period.

12.  Line 103-104, this information belongs in the methods section. 

13.  A second map that uses proportional symbols to show emissions by compressor would be quite interesting. It would also allow us to see if there is a spatial pattern in (1) emissions and (2) missing data on emissions. 

14.  Table 2 should make it clearer that benzene, formaldehyde etc., are a subset of VOCs. Perhaps re-label “Rank” as “Rank by pounds released.” Some of these chemicals are much more hazardous at a lower level of exposure, so total pounds might not be the most important characteristic.

15.  Please specify how many compressor stations worth of data is included in Table 2 in the table footnote. Ditto for Tables 3 and 4. Also, relabel Table 3 USEPA NEI Compressor Station Emissions (Pounds)... How many compressors are there in each of these towns? A second column with this information would be helpful.

16.  In general, all tables need more information included in the caption. They should stand alone. What is the source of the data for each? What is the temporal resolution? How many compressors are featured?

17.  Line 132: please add a citation. In addition, this coal paragraph is very U.S.-centric. While coal-fired power plants have gone offline in the US, many other countries are still building them at a rapid pace. 

18.  Line 136: requires a citation.

19.  Another point of potential interest related to the contribution to greenhouse gases is that cheap natural gas may be diverting efforts from renewable development (see Saunders 2016 Environ Gechem Health) 

20.  Line 150-151: Please rephrase, “Air pollution from whatever source is being documented as being a risk factor for many different diseases.”   

21.  Line 162: For the Finkel paper, please note that they did not observe cancer incidence above expectation for leukemia or thyroid cancer.

Author Response

Our analysis was only for emissions from natural gas, which is the primary fuel at compressor stations.  Diesel is usually available as a backup fuel.  But the National Emissions Inventory only reports natural gas as the fuel source at compressor stations, so we have no information on diesel emissions if there ever are any from compressor stations.. 

Round 2

Reviewer 1 Report

The authors did nothing to address my concerns. In their response, they say monitoring data or air dispersion modeling would improve the paper, yet they don't mention it at all as a limitation of the study in the paper.  At a minimum, they could calculate what fraction of emissions these Title V permits are in comparison to all the Title V permits in New York.  That would cost them nothing and be easy to do.

Author Response

Air Modeling: It is our understanding that an integral part of the Title V permit process is the state's review of modeling data prepared by the company seeking a permit. This model must be based on EPA-approved methodologies. It is our further understanding that if the state finds a company's submission inadequate in some respect, it either (a) requests that it be re-done taking into account their observations or (b) that the state would create it's own model. The grant of a permit indicates that the air dispersion model is accurate and that discharges are within federal and state regulatory limits, including those specifically related to Title V facility. The authors have no indication that the modeling used to permit a specific compressor station or compressor station as a group is inadequate or flawed. However, we believe there is a weakness in the existing approach to modeling which considers each chemical individually and fails to consider the effects of continually potential exposure to multiple chemicals at any given time. This problem is not one that monitoring per se would address, and there is no regulatory requirement that the health effects of simultaneous exposures be considered before the granting of a permit. What monitoring would provide is an actual record of the fluctuation in emissions over time (whereas modeling essentially assumes a constant rate of emissions; meteorological data is the primary dynamic part of the equation). Give that there are hundreds of Title V facilities in any given state, the cost of monitoring even a small percentage is considerable. Our observation is that the state chooses to spot monitor a plant only when they have reason to believe there is a possible violation.

Comparison to all Title V permitted facilities. We agree it would be a valuable exercise to compare compressor stations to all Title V facilities. And, in fact, we attempted to do just that. However, we could not locate a NYS or a federal list of all Title V facilities. And, if we had, it would not have been a trivial matter to determine the releases for this large set of facilities.     

Hope that clarifies our thinking.